# Molecular Traceability Approach to Assess the Geographical Origin of Commercial Extra Virgin Olive Oil

**DOI:** 10.3390/foods13142240

**Published:** 2024-07-16

**Authors:** Michele Antonio Savoia, Isabella Mascio, Monica Marilena Miazzi, Claudio De Giovanni, Fabio Grillo Spina, Stefania Carpino, Valentina Fanelli, Cinzia Montemurro

**Affiliations:** 1Department of Soil, Plant and Food Sciences, University of Bari Aldo Moro, Via Amendola 165/A, 70126 Bari, Italy; michele.savoia@uniba.it (M.A.S.); isabella.mascio@uniba.it (I.M.); monicamarilena.miazzi@uniba.it (M.M.M.); claudio.degiovanni@uniba.it (C.D.G.); cinzia.montemurro@uniba.it (C.M.); 2Department of the Central Inspectorate for the Protection of the Quality and Repression of Fraud of Food Products (ICQRF), Via Quintino Sella 42, 00187 Roma, Italy; fabio.grillospina@gmail.com; 3Central Inspectorate for Fraud Repression and Quality Protection of the Agrifood Products and Food (ICQRF), The Ministry of Agriculture, Food Sovereignty and Forests (MASAF), Via Quintino Sella 42, 00187 Roma, Italy; s.carpino@masaf.gov.it; 4Spin Off Sinagri s.r.l., University of Bari Aldo Moro, Via Amendola 165/A, 70126 Bari, Italy

**Keywords:** molecular traceability, EVOO, olive, molecular markers, SSR, private alleles

## Abstract

Extra virgin olive oil (EVOO) is a precious and healthy ingredient of Mediterranean cuisine. Due to its high nutritional value, the interest of consumers in the composition of EVOO is constantly increasing, making it a product particularly exposed to fraud. Therefore, there is a need to properly valorize high-quality EVOO and protect it from fraudulent manipulations to safeguard consumer choices. In our study, we used a straightforward and easy method to assess the molecular traceability of 28 commercial EVOO samples based on the use of SSR molecular markers. A lack of correspondence between the declared origin of the samples and the actual origin of the detected varieties was observed, suggesting possible adulteration. This result was supported by the identification of private alleles based on a large collection of national and international olive varieties and the search for them in the molecular profile of the analyzed samples. We demonstrated that the proposed method is a rapid and straightforward approach for identifying the composition of an oil sample and verifying the correspondence between the origin of olives declared on the label and that of the actual detected varieties, allowing the detection of possible adulterations.

## 1. Introduction

Extra virgin olive oil (EVOO) is one of the most important ingredients for flavoring and cooking food in Mediterranean cuisine. EVOO has high nutritional value and is rich in antioxidant properties due to the abundant presence of unsaturated fatty acids and some bioactive components, such as phenolic compounds, tocopherols, and carotenoids [1]. Therefore, the consumption of EVOO is associated with a reduced risk of cardiovascular disease as well as prevention of cancer and type 2 diabetes [2,3]. Previous studies showed how several factors such as environmental conditions, the phenological phase of the plant, and storage conditions [4,5,6,7,8,9] influence the MUFA, PUFA, and polyphenol contents.

These factors, along with the olive cultivars used, allow for the attainment of high levels of total polyphenols and other bioactive compounds, resulting in the production of a high-value EVOO with distinctive sensorial characteristics, such as bitterness, pungency, and fruitiness intensity [10]. 

We are witnessing the increasing interest of health-conscious consumers in the composition of the food they buy and the growing market of oils with geographical indication. Therefore, there is a need to properly valorize high-quality EVOO and protect it from fraudulent manipulations to safeguard consumer choices. The high economic interest related to the certified EVOO market has led to an increase in fraudulent practices, which mainly consist in the addition of cheaper oils from cultivars or countries not listed on the label or reported in the product disciplinaries [11].

Commonly, each Mediterranean country uses national cultivars to produce olive oils. This allows the easy identification of the country of origin of olives used to produce a specific EVOO through the detection of cultivars present in the oil. However, due to the spread of intensive agronomic systems, some cultivars are considered widespread “global varieties” [12]. These could thwart the identification of an olive oil’s origin and the subsequent verification of its compliance with its labeling.

Traditionally, to detect adulteration and ensure the traceability of EVOO, analytical approaches based on the evaluation of chemical compounds were used [13,14,15]. However, they have been proven to be strongly affected by environmental conditions and would require a database that includes both chromatographic and spectroscopic information from the most relevant cultivars [16]. For these reasons, the application of DNA-based methods stands out as a prime technique. These approaches are suitable for the analysis of highly processed food matrices and have been shown to be highly reproducible and reliable, being unaffected by environmental conditions. Their efficient use in varietal identification and adulterant detection in olive oil has been widely demonstrated [17,18,19].

It is well known that consumers are willing to pay higher prices for EVOOs from Italy, which is a country with a long and established olive-growing tradition [20]. Indeed, it was demonstrated that origin and territory as well as taste and nutritional information were attributes that significantly affected olive oils’ prices [21]. Therefore, DNA-based methods able to identify the territorial origin of olive oils become a key element for consumer protection.

Over the years, several molecular markers have been used for varietal identification in EVOOs. Early approaches were based on AFLP and RAPD molecular markers. AFLP markers combine restriction enzymes with polymerase chain reaction (PCR). These markers ensure good reproducibility and a high degree of polymorphism. However, the complexity of the olive oil matrix limits the reliability of the obtained profiles [22,23]. For RAPD markers, genomic DNA is amplified with short random primers [24]. They are simple to analyze and applicable even to genetically unknown species but are poorly reproducible, so they are usually used in combination with other molecular markers [25,26].

To date, the most commonly used molecular markers are SSRs and SNPs [27], showing high efficiency in highly fragmented DNA.

Thanks to the advent of next-generation sequencing (NGS) techniques, SNP markers have been widely used in olive oil traceability [18,28]. These markers are abundant and distributed in the entire genome and their detection is highly reproducible [29]. Additionally, the genotyping by sequencing (GBS) [30] technique provides thousands of SNP markers useful for assessing the authenticity of table olives and oil products [18]. Further, the high-resolution melting (HRM) approach can be used for the detection of SSR and SNP markers [19,31,32,33]. In addition, for traceability purposes of EVO oils, other AFLP- or RAPD-derived markers such as sequence-tagged sites (STSs) [26] have been used. However, simple sequence repeat (SSR) molecular markers, due to their high reproducibility, elevated polymorphism degree, standardized and straightforward detection systems, and cost-effectiveness, are the most commonly used for discrimination analyses of plant varieties and food traceability purposes [26,34,35].

For a considerable number of varieties widely used for EVOO production, the SSR profiles are currently available, thus making the use of SSR molecular markers a powerful tool in EVOO traceability [36,37]. However, in the case of olive oils consisting of several varieties (blends), molecular analysis can often provide a high number of alleles attributable to both European and non-European cultivars. In this case, the use of private alleles has proven to be a suitable strategy to reduce the allelic complexity and identify the geographical macro-areas of origin of the cultivars used to produce the oils [38,39,40].

A previous study performed by Piarulli et al. [41] developed an efficient and highly reproducible protocol to isolate whole genomic DNA from commercial filtrated EVOOs suitable for traceability purposes. In our study, we validated this approach by analyzing 28 commercial EVOOs through nine olive-specific SSR markers with the aim of verifying the correspondence between the origin of the varieties detected through the molecular analysis and the information indicated on the product label. The comparison of the EVOOs’ molecular profiles with those of 149 international olive accessions allowed to investigate the genetic similarities existing between the analyzed samples and the reference cultivars. In addition, a list of private alleles was produced in order to associate each allele to a specific country and verify the declared geographical origin of the olive.

## 2. Materials and Methods

### 2.1. EVOO Samples

Twenty-eight EVOO samples were provided from the ICQRF (Central Inspectorate of Protection of Quality and Fraud Repression for agricultural and food). Among these, 19 samples were commercial bottles with labels while the remaining 9 were not yet bottled (Appendix A). For the commercial bottles, information about the composition was present on the label, while for the remaining samples, this information was provided by the producer. In addition, a monovarietal oil obtained from the Leccino variety was used as a quality control sample (named QC).

### 2.2. SSR-Based Assay

DNA was isolated from the oil samples following the method of Piarulli et al. [41], and DNA quantity and quality were assessed using a Nano-Drop™2000C Spectrophotometer (Thermo Scientific, Waltham, MA, USA) and 1% agarose gel electrophoresis. The samples were molecularly characterized using a set of nine simple sequence repeat (SSR) markers widely used in olive genotyping: DCA05, DCA17, DCA18, DCA15, EMO90, EMOL, DCA09, DCA03, and GAPU101 [42,43,44]. The polymerase chain reactions (PCRs) were performed following the protocol described by Piarulli et al. [41]. The PCR products were visualized on 1.8% agarose gel and the amplicons were further analyzed through capillary electrophoresis using an ABI PRISM 3100 Genetic Analyzer from Life Technologies (Carlsbad, CA, USA).

### 2.3. Data Analysis

The allele sizes of the 28 analyzed samples were assessed through the GeneMapper Software, version 3.7 (Life Technologies, Carlsbad, CA, USA). In order to study the genetic relationships of the 28 oils, a gene pool of 149 olive cultivars widespread in the Mediterranean basin and widely used for oil production was used as a reference (dataset provided by the Department of Soil, Plant, and Food Sciences, Di.S.S.P.A.—University of Bari). In detail, the reference dataset consists of Italian (64), Tunisian (26), European Union (French, Greek, Spanish, and Portuguese) (23), and non-European Union (Albanian, Algerian, Lebanese, and Kurdish) (36) cultivars.

The informativeness of the chosen SSRs was assessed by calculating the polymorphism information content (PIC) value using the reference varieties through Cervus version 3.0 [45]. To assess the varietal composition of the oil samples, a comparison between the alleles detected in the oils and those of the reference list of cultivars was performed. In particular, each allele found in the oil samples was searched in the reference molecular profiles. Moreover, to analyze the genetic relationships between the oil samples and the reference varieties, a neighbor-joining (NJ) analysis and a principal coordinates analysis (PCoA) [46] were performed using DARWIN v. 6.0.010 (http://darwin.cirad.fr) [47] and GenAlEx v.6.5 [48] software, respectively. The molecular profiles of the olive cultivars were further assayed for private alleles (allelic frequency < 1%) [49] using GenAlEx v.6.5 software.

## 3. Results

### 3.1. EVOO Sample Genotyping by SSR Markers

DNA was successfully extracted from all the samples. A partial DNA degradation was observed on agarose gel, and the amount of recovered DNA was between 12 and 32 ng/µL. PCRs were performed using primer couples specific for nine olive SSR markers (Appendix A). Polymorphism information content (PIC) values showed a high level of informativeness of the chosen SSR markers as the PIC values were equal to or higher than 0.68 for all the tested loci (Table 1). In particular, the most informative markers were DCA03 and DCA09, showing PIC values of 0.87 and 0.89, respectively.

All oil samples provided a clear and distinguishable pattern of amplification for most of the analyzed microsatellite loci. The molecular profile of the quality control sample showed total correspondence with that of the Leccino variety, supporting the high reliability of the approach (Table 2). Unfortunately, amplification was not obtained for all assayed SSR markers (e.g., DCA05). For most of the tested SSR markers, the presence of multiple alleles was detected, indicating that the mixtures were composed of more than one cultivar. Samples OT8 and OT28 showed the highest number of total alleles (18).

### 3.2. Genetic Similarities with the Reference List of Varieties

To attempt identification of the varietal composition and geographical origin of the 28 EVOO samples, we used a step-by-step approach based on the comparison of the oil samples’ molecular profiles with those of the reference cultivars. First, we searched for the presence of all the possible allelic combinations detected in the oils by comparison with the reference list of cultivars and profiles in order to obtain an idea of the samples’ varietal composition based on the country of origin (Figure 1). Each allele combination found in the oil samples (considering the diploid nature of olive) was searched in the reference molecular profiles. For all samples, allele combinations of varieties from Italy, Tunisia, and European Union and non-EU countries were found, suggesting the mixed composition of the oils. In particular, the samples OT9, OT15, and OT18 showed the highest percentage of Italian allelic combinations (60%, 64%, and 65%, respectively). The highest percentage of Tunisian allelic combinations was observed in sample OT12 (31%). The samples OT5 and OT23 showed the highest percentages of European Union allelic combinations (29% and 31%, respectively). Finally, the highest percentage of non-EU cultivar allelic combinations was found in sample OT19 (35%).

To further investigate the identity of varieties possibly used in the production of the EVOO samples, the second step was to construct a tree through the unweighted neighbor-joining (UWNJ) method (Figure 2) and perform a principal coordinates analysis (PCoA) (Figure 3).

The phylogenetic analysis allowed the identification of three groups of interest. Group 1 consists of two subgroups: 1A, including most of the analyzed EVOO samples (25), and 1B, comprising the sample OT10 and the Italian cultivar Ogliarola Salentina. The presence of most of the analyzed samples in the same subgroup supports the hypothesis that some varieties are shared among all the oils, although the varietal composition declared on the product label is quite different for each sample. Group 2 includes the sample OT27 and the Italian cultivars Gentile di Chieti, Cassanese, and Cellina di Nardò. Finally, Group 3 comprises sample OT19 along with some Italian, Albanian, Tunisian, and Lebanese cultivars, confirming the presence of non-EU cultivars in the allelic combinations found in Figure 1.

The principal coordinate analysis (Figure 3) confirmed the phylogenetic clustering, grouping together almost all of the 28 oil samples into two groups. In Group A, located in the upper right quadrant of the graph, most of the oil samples are present, along with some of the Italian varieties and the Tunisian variety Neb Jemal 1. The samples OT10 and ITA_Ogliarolasalentina (in yellow in Figure 3) and OT27 and ITA_Gentiledichieti (in green in Figure 3), belonging to Group 1B and Group 2, respectively, in the NJ analysis, had their genetic relation confirmed in the PCoA. As observed in the NJ analysis, the sample OT19 was found to be separate from the other oil samples (lower right quadrant of the graph) and was grouped with Algerian, Italian, Tunisian, and Albanian cultivars (Group B). In this group, the cultivars ITA_DolceAgogia and TUN_Gerboui (in blue in Figure 3) are also present, being part of Group 3 in the NJ dendrogram.

### 3.3. Private Allele Identification

The last step was a search for private alleles (allelic frequency < 1%) distinctive of the different gene pools and countries, which was performed using the dataset of 149 olive cultivars (Table 3). Next, we searched for the presence of these private alleles in the 28 analyzed samples (Table 4).

The private allele analysis in the reference olive population showed the presence of private alleles belonging to Italian, French, Spanish, Portuguese, Algerian, Tunisian, Kurdish, and Lebanese germplasm, while no private alleles belonging to Greek and Albanian germplasm were detected. The marker DCA17 proved to be the most informative in terms of private alleles detected since a private allele was found for five different countries.

The search of private alleles showed their presence in almost all the samples. In particular, Italian, Algerian, and Tunisian private alleles were identified. In samples OT2, OT10, OT11, OT13, OT19, OT26, OT27 and OT28, the country associated with the allele matched the declared olive origin. The sample OT5 showed the presence of an Algerian private allele, although the declared origin of the olives was Tunisia. The sample OT9, for which the declared origin of the olives was the EU, presented an Algerian private allele along with an Italian one. The sample OT12 showed the presence of an Algerian private allele, although the declared origin of the olives was Italy, Spain, and Greece (EU origin). The samples OT15, OT18, and OT23, declared as being produced with Italian olives, presented Algerian (OT15 and OT18) and Tunisian (OT23) private alleles.

## 4. Discussion

### 4.1. Use of SSR Markers for Traceability of Olive Oil

One of the major concerns of consumers is about the origin and safety of the food products they buy. Food traceability and authentication are central tools to reassure consumers about food transparency and safety. Recently, the demand for molecular tools for food authentication and traceability has significantly increased, especially for the purpose of protecting and properly valorizing high-quality food products. Although the number of studies based on the use of SSRs for agri-food traceability and authentication has gradually decreased over the past decade, along with an increase in the number of works employing more abundant and stable SNP markers, microsatellite markers remain the most widely used markers for molecular traceability [34]. In particular, the detection of private microsatellite alleles in olive oils can verify the geographical origin on the label, as proven before [38,39,40]. Their use protects both the consumer and the producer, justifying the EVOO prices related to the geographical origin of the cultivars.

In this study, we analyzed 28 commercial EVOOs provided by the ICQRF (Central Inspectorate of Protection of Quality and Fraud Repression for agricultural and food) with the aim of verifying the correspondence between the geographical origin declared on the product label and the origin of the varieties putatively present in the samples as identified through molecular analysis. Isolation of DNA from agri-food samples is a crucial step in the molecular traceability process. Most processed food products undergo physical treatments affecting DNA quality. Moreover, the presence of inhibitors in processed foods can prevent downstream molecular applications [50]. For DNA extraction from the commercial samples, we used the well-established protocol of Piarulli et al. [41]. DNA was successfully isolated from all the samples, corroborating the effectiveness of the protocol. The DNA samples were amplified with nine primer couples specific for olive SSR markers. The microsatellites were selected for their high discriminatory power and reproducibility, attributed to their high polymorphism, easily scored patterns, and small-scale stuttering [37,41,51]. Their informativeness was demonstrated by the obtained PIC values (Table 1). Despite the discreet quality of isolated DNA, the amplification product was not obtained for all the assayed markers; in particular, the marker DCA05 provided the worst results, with only seven successfully amplified samples. The complexity of the analyzed food matrix led to a high degradation level of the extracted DNA, which may have caused the different amplifiability of each sample and diverse performance of each marker [51]. However, the complete correspondence between the molecular profile of the quality control sample and that of the Leccino variety supports the reliability of the obtained molecular profiles.

For the amplified samples, a clear amplification profile was obtained. In some cases, more than two alleles, as expected for a monovarietal oil, were detected, indicating the presence of more than one variety in the oil samples. The efficiency of SSR markers in the monovarietal olive oil traceability process has been widely demonstrated [17,37,51,52]; however, only a few studies describe their application in commercial samples [53,54].

### 4.2. Varietal Composition Analysis of the EVOO Samples

The origins of the varieties used to produce the EVOO samples, as declared by the producers, were different for each EVOO and included Italian, Tunisian, Spanish, Greek, and other EU and non-EU cultivars. We attempted to identify the varieties putatively used in the oil production, and therefore their geographical origin, through comparison of their molecular profiles with those of 149 international accessions commonly used for EVOO production. The use of molecular markers for verification of the geographical origin of olive oils is considered a reliable approach [16,25].

First, we searched for the recurrence of the allelic combinations found in the samples in the reference varieties. As expected, for all the oils, allele combinations of varieties from Italy, Tunisia, and European Union and non-EU countries were found, indicating the use of varieties from different countries. The allelic composition of samples OT5, OT12, OT19, and OT23 was particularly noteworthy. Samples OT5 and OT23 presented the highest percentage of European Union allelic combinations, although the declared origins of their olives were Tunisia and Italy, respectively. Sample OT12, declared as being produced with Italian, Spanish, and Greek olives, showed the highest percentage of Tunisian allelic combinations. Finally, OT19 presented the highest amount of non-EU cultivar allelic combinations, although the declared origin of the olives was Italy. Recently, in Italy, an increase in the price of EVOO of domestic origin has been observed, with a negative impact on the EVOO market [55]. Therefore, some producers use a blend of European or non-EU olives to produce EVOO passed off as 100% Italian [56]. The presence of varieties originating from diverse countries, especially in samples declared as being composed of 100% Italian olives, is not surprising considering that mislabeling and substitutions represent the most common EVOO frauds [16].

The deepest investigation of the composition of the oils was performed through the construction of a tree through the UWNJ method and using a PCoA. They showed similar results, indicating a shared genetic background of the EVOO samples and suggesting that the same varieties were used to produce all the oils. The correspondence between the two approaches of tree construction and PCoA was also observed before [41], indicating the high reliability of these two methods of analysis.

### 4.3. Private Allele Identification

Private alleles are alleles that are present only in a single population among a collection of populations. They allow the identification of samples belonging to a specific population located in a certain geographical area [57]. These alleles have proven to be informative for diverse types of population genetic studies, including discrimination between olive cultivars and their assignment to a geographically defined population [38,40,58]. The use of private alleles has been revealed to be highly efficient in the identification of different gene pools present in the populations of different plant species, such as almond, grapevine, and cocoa [59,60].

In order to more accurately identify the origin of the olives, a search for private alleles was performed. The marker DCA17 emerged as the most informative out of the private alleles detected, as demonstrated in other studies [38,61]. Private alleles of Italian varieties were detected in almost all samples, including those indicated as being produced using Tunisian olives. Algerian private alleles were also detected in 12 oils, including OT15, OT18, and OT25, whose declared olive origin was Italy, and OT9, OT12, and OT20, whose declared olive origin was EU countries. In samples OT8, OT23, and OT25, declared as being produced exclusively with Italian olives, a Tunisian private allele was found. Interestingly, sample OT19 (declared as being produced with Italian olives), belonging to a phylogenetic group including EU and non-EU varieties, showed only an Italian private allele.

Theoretically, in order to have an exhaustive list of a country’s private alleles, it would be necessary to analyze the molecular profiles of all the varieties of that country. However, by considering a sufficiently large number of accessions, it is possible to have a fairly accurate private alleles list. In our case, we considered 149 genotypes; therefore, for some of the considered countries, we did not have an adequate number of private alleles. An exception may be represented by the panel of Italian accessions, including 64 widespread olive cultivars, for which the identified list of private alleles could be considered reliable. Expansion of the number of accessions from the other countries will allow for improvement of the method of identification of the geographical origin of accessions used in olive oil production. The discriminating power and usefulness of private alleles (SSRs or SNPs) for tracing the origin of foods have been widely proven in other food chains such as pasta [62,63].

The reported method represents a first approach for the investigation of the origin of commercial EVOOs through molecular identification of the varietal composition. Half of the olive oils labeled as “Italian” showed alleles of cultivars from non-EU countries. All three samples that reported the origin as “European Union” presented extra alleles typical of non-EU countries, and almost 50% of the samples labelled as “Italian, Greek, and Spanish” showed non-EU alleles. This kind of approach, if combined with documental traceability and antifraud operations, offers the possibility to acquire additional information on the varietal composition and, indirectly, on the origin of the raw material.

## 5. Conclusions

Our work aimed to present a molecular strategy useful to authenticate commercial EVOOs and identify the varieties putatively used in the production process. Moreover, the identification of private alleles, based on a large collection of national and international olive varieties, was demonstrated to be useful to verify the geographical origin of the olives used for oil production, allowing the detection of false information declared on the product labels. Although this approach is not quantitative but only qualitative, it may represent a straightforward and easy method for the preliminary identification of the origin of olives used to produce EVOOs and the detection of undeclared varieties, allowing the discovery of any possible adulterations. In the future, an increase in the number of international olive varieties assessed will allow to refine the list of private alleles and set up a systematic approach for identification of the geographic origin of cultivars used in the production of olive oils, allowing the valorization and protection of high-quality products, such as certified EVOOs.

## Figures and Tables

**Figure 1 foods-13-02240-f001:**
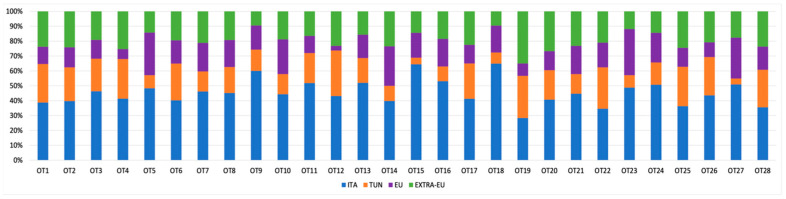
Bar plot representing the proportion of allelic combinations for each of the 28 samples.

**Figure 2 foods-13-02240-f002:**
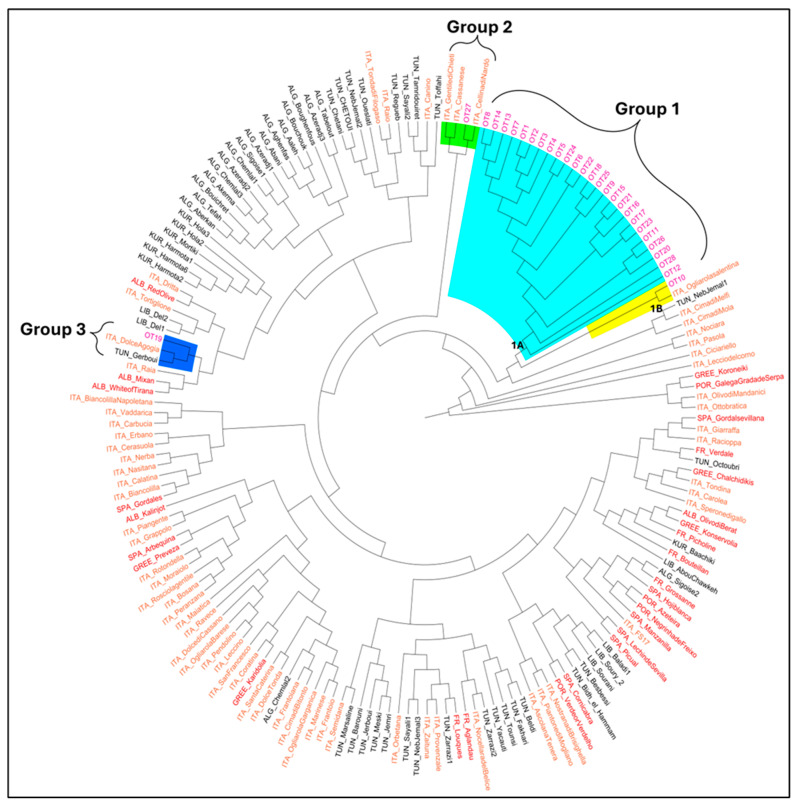
Genetic similarity between the 28 oil samples and the subset of 149 olive cultivars based on genetic distance. The oil samples are indicated in pink, the Italian cultivars in orange, the EU cultivars in red, and the non-EU cultivars in black. The groups to which the oil samples belong are highlighted in light blue (Group 1A), yellow (Group 1B), green (Group 2), and blue (Group 3).

**Figure 3 foods-13-02240-f003:**
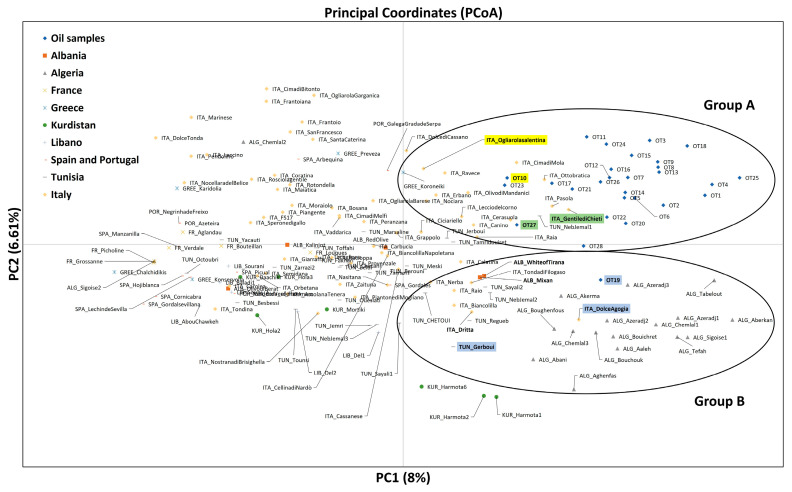
Principal coordinates analysis (PCoA) performed on the 28 oil samples and the set of 149 Mediterranean olive accessions. The samples and the accessions close to each other in the phylogenetic analysis are highlighted in different colors.

**Table 1 foods-13-02240-t001:** PIC values obtained for the nine microsatellite loci assayed.

Locus	PIC
DCA03	0.87
DCA05	0.75
DCA09	0.89
DCA15	0.68
DCA17	0.77
DCA18	0.83
GAPU101	0.84
EMO90	0.71
EMOL	0.69
Mean value	0.78

**Table 2 foods-13-02240-t002:** Allelic profiles expressed in base pairs (bp) obtained for the 28 EVOO samples. Sample QC is the quality control sample represented by the Leccino monovarietal oil. na = not available.

Sample Code	DCA03	DCA05	DCA09	DCA15	DCA17	DCA18	GAPU101	EMO90	EMOL
OT1	232-235	na	174-180	246-257	111	169-171-173-175-177	182-200	194	na
OT2	232-235	na	182	246	na	169-171-173-175-177	182-200	188-192	na
OT3	232-237	na	180-182	246	na	169-171-173-175-177	182-198-216	188	196
OT4	232-237	192	172-174	246	109-111	169-171-175	182-198-216	190	192
OT5	na	na	174-180	na	na	169-171-175	182	188-190	196-198
OT6	237-241	na	174-176	246	113	169-171-173-175-177	182-196	na	198
OT7	235-237	na	174	257	na	169-171-177	182	188	na
OT8	232-235-237-247	na	na	257	113	169-171-173-175-177-179	182-214	188-192	194-198
OT9	232-239-241	na	172-174	na	na	175	182-198	188-194-196	190
OT10	232-239	na	na	257	na	167-175	182-198	188	192-198
OT11	237	na	172	246	na	175-177-179	182-198	188-194	na
OT12	232-243-245-247	na	180-182	246	na	175-177	182	194	na
OT13	237	na	172-178	257	113	169-171-173-175-177	182-200	188	210-212
OT14	237-241-247	na	na	257	na	169-171-175	182-218	188-190-192	198
OT15	235-237	na	172	na	na	167-169-173	182	186-188-194	206-208
OT16	232-237	na	na	na	na	169-177	182-198	186-188-190	na
OT17	237-247	na	na	246	na	na	182-198-210-214	188	na
OT18	229-232	194-200-202	172-174-180	257	na	169-175	182-198	188-194	210
OT19	na	194	174	246	na	171-173	190-216	186	208
OT20	204	na	175	246	188-192	208-210-212	174-180	na	198
OT21	na	107	169	246	186-188-190-192-194	198	174	237	182
OT22	194-202	111	169-171-175-177	246	196	198	174	229-239	182
OT23	na	107	na	257	188-190	198-212	162	237	182-214-216
OT24	-	-	169-171-173-175	246	188-190-194	196-198	172	235-241-247	182-198
OT25	194-202	-	171-173-175	246	194-196	206-210-212	172-174	232-237	182-214
OT26	-	109-113	175-177	246	188	208-210	-	229-235-237	182
OT27	-	-	171-173	-	188-190	210-212	-	243	192-196
OT28	204-206	-	171	246-257	196	196-204-208-210-212	-	232-235-237-245	182-200-218
QC	243-253	198-206	162-162	246-266	107-117	177-177	198-200	188-194	198-198

**Table 3 foods-13-02240-t003:** Private alleles detected in the reference olive dataset (Di.S.S.P.A.).

Country	Locus	Allele	Freq %
Algeria	*DCA03*	241	0.167
		251	0.194
	*DCA09*	164	0.167
		180	0.028
		196	0.083
	*DCA15*	259	0.222
		268	0.250
	*DCA17*	123	0.036
	*GAPU101*	180	0.056
	*EMOL*	202	0.222
		206	0.028
France	*DCA17*	169	0.083
Kurdistan	*DCA15*	248	0.214
	*EMO90*	200	0.357
Lebanon	*DCA17*	159	0.167
Spain and Portugal	*DCA03*	227	0.042
Tunisia	*DCA15*	270	0.058
	*DCA17*	153	0.020
		177	0.020
	*DCA18*	191	0.019
	*GAPU101*	214	0.019
Italy	*DCA03*	235	0.008
		257	0.008
	*DCA05*	196	0.008
		212	0.030
	*DCA09*	166	0.030
		200	0.008
		208	0.008
	*DCA15*	251	0.008
		258	0.015
		273	0.008
	*DCA17*	127	0.015
		129	0.023
		161	0.015
		165	0.008
		175	0.031
		183	0.008
		189	0.008
	*DCA18*	159	0.015
		201	0.008
	*GAPU101*	190	0.040
		192	0.048
		196	0.008
		198	0.048
		200	0.040
		218	0.008
	*EMOL*	204	0.019
		212	0.010

**Table 4 foods-13-02240-t004:** Private alleles detected in oil samples for the nine microsatellite loci assayed. The presence of a private allele specific to a country is indicated with the symbol “x” for each sample.

Sample Code	Declared Origin of the Olives	Algeria	France	Kurdistan	Lebanon	Spain and Portugal	Tunisia	Italy
OT1	Tunisia	x						x
OT2	Italy							x
OT3	Tunisia	x						x
OT4	Spain							x
OT5	Tunisia	x						
OT6	Tunisia	x						x
OT7	Tunisia							x
OT8	Italy						x	x
OT9	EU	x						x
OT10	EU							x
OT11	Italy, Spain, Greece, and Tunisia							x
OT12	Italy, Spain, and Greece	x						
OT13	Italy, Spain, Greece, and Tunisia							x
OT14	Italy, Spain, Greece, and Tunisia	x						x
OT15	Italy	x						x
OT16	EU and non-EU							x
OT17	Tunisia						x	x
OT18	Italy	x						x
OT19	Italy							x
OT20	EU	x						x
OT21	EU and non-EU							x
OT22	Italy							
OT23	Italy						x	x
OT24	EU and non-EU	x						x
OT25	Italy	x					x	x
OT26	Italy, Spain, Greece, and Tunisia							x
OT27	Italy, Spain, Greece, and Tunisia							x
OT28	EU and non-EU							x

## Data Availability

The original contributions presented in the study are included in the article/Appendix A, further inquiries can be directed to the corresponding author.

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
