# Peer review of "Molecular Traceability Approach to Assess the Geographical Origin of Commercial Extra Virgin Olive Oil"

_foods, 2024, doi:10.3390/foods13142240_

Round 1
Reviewer 1 Report
Comments and Suggestions for Authors
The manuscript offers a significant contribution to the field of food traceability, particularly concerning the geographical origin of Extra Virgin Olive Oil (EVOO). The utilization of Simple Sequence Repeats (SSR) molecular markers represents a robust scientific strategy, and the study's findings are relevant for addressing fraudulent activities within the olive oil industry. Thus, I recommend major revision since there are several points that need improvement before the publication in Foods. The issues are briefly described as follows:
1. The term "Extra-virgin olive oil (EVOO)" should have consistent formatting throughout the manuscript. The hyphen should be removed, and it should be consistently written as "Extra Virgin Olive Oil (EVOO)." The abbreviation "EVO oils" should be corrected to "EVOO" to maintain consistency
2. In Table 1, the phrase "The declared origin of olives 120 used for oil production is also indicated.”should be moved to the main text or a footnote.
3. Line 217, the word "separated" should be used instead of "separate" to maintain the correct verb tense.
4. The manuscript should include electrophoresis images of the PCR products as supplementary material to enhance the transparency and reproducibility of the results.
5. The analysis of the results in Table 3 requires more detail. Additionally, the explanation for the lack of amplification of the DCA05 marker is not convincing and should be addressed more thoroughly.
6. The paper should present the results of laboratory quality control samples to validate the accuracy of the method.
7. In Table 5, the meaning of "x" should be explained in a footnote for clarity.
8. The manuscript should include information on the application of other molecular markers in the authentication of EVOO, as well as a comparison of the advantages and disadvantages of SSR markers relative to other types of markers.
Comments on the Quality of English LanguageThe English in this manuscript is overall clear and grammatically correct.
Author Response
The manuscript offers a significant contribution to the field of food traceability, particularly concerning the geographical origin of Extra Virgin Olive Oil (EVOO). The utilization of Simple Sequence Repeats (SSR) molecular markers represents a robust scientific strategy, and the study's findings are relevant for addressing fraudulent activities within the olive oil industry. Thus, I recommend major revision since there are several points that need improvement before the publication in Foods. The issues are briefly described as follows:
We would thank the reviewer for the careful reading of the manuscript and considerations. We addressed all the points in detail below.
- The term "Extra-virgin olive oil (EVOO)" should have consistent formatting throughout the manuscript. The hyphen should be removed, and it should be consistently written as "Extra Virgin Olive Oil (EVOO)." The abbreviation "EVO oils" should be corrected to "EVOO" to maintain consistency
We modified the text as suggested by the reviewer.
- In Table 1, the phrase "The declared origin of olives 120 used for oil production is also indicated.” should be moved to the main text or a footnote.
We modified it (now table S1).
- Line 217, the word "separated" should be used instead of "separate" to maintain the correct verb tense.
We corrected it.
4. The manuscript should include electrophoresis images of the PCR products as supplementary material to enhance the transparency and reproducibility of the results.
We included the picture of agarose gel electrophoresis of PCR products obtained with primers corresponding to the DCA18 marker as supplementary figure S1.
5. The analysis of the results in Table 3 requires more detail. Additionally, the explanation for the lack of amplification of the DCA05 marker is not convincing and should be addressed more thoroughly.
We improved the analysis and the discussion of results shown in Table 3 explaining more deeply and comprehensively the lack of amplification for some markers.
6. The paper should present the results of laboratory quality control samples to validate the accuracy of the method.
We included the results of the laboratory quality control sample (represented by the monovarietal olive oil obtained using the variety Leccino) in Table 3. Moreover, the amplification of this control with primers corresponding to the DCA18 marker is shown in supplementary figure S1.
7. In Table 5, the meaning of "x" should be explained in a footnote for clarity.
We added it.
8. The manuscript should include information on the application of other molecular markers in the authentication of EVOO, as well as a comparison of the advantages and disadvantages of SSR markers relative to other types of markers.
We included information about the use of different molecular markers in the authentication of EVOO in the Introduction section.

Reviewer 2 Report
Comments and Suggestions for Authors
1Dear Authors, the following points need to be addressed:
11. Abstract is too general, and key quantitative results should be included in the abstract.
22. Line 86: if you mentioned “Countries”, should be more than one, not only Italy.
33. Table 1 should be moved in a supplementary table, as the same information regarding declaration of olives origin is repeated in Table 5.
44. “Discussion” section is very poor, there are no comparisons with other values ​​from scientific articles, only general information that is more like an “Introduction” part.
55. A final line describing the future outlook on this work should be included in the conclusion.
Author Response
Dear Authors, the following points need to be addressed:
We would thank the reviewer for the careful reading of the manuscript and considerations. We addressed all the points in detail below.
1. Abstract is too general, and key quantitative results should be included in the abstract.
We improved the abstract as suggested by the reviewer.
2. Line 86: if you mentioned “Countries”, should be more than one, not only Italy.
We modified it.
3. Table 1 should be moved in a supplementary table, as the same information regarding declaration of olives origin is repeated in Table 5.
We moved the Table 1 in the Supplementary materials.
4. “Discussion” section is very poor, there are no comparisons with other values from scientific articles, only general information that is more like an “Introduction” part.
We improved the Discussion session as suggested by the reviewer. However, it should be noted that the number of studies analyzing commercial olive oils is limited reducing the chances of making comparisons.
5. A final line describing the future outlook on this work should be included in the conclusion.
We included the future outlook in the conclusion section.
Reviewer 3 Report
Comments and Suggestions for Authors
the EVOOs were the food and only the SSR markers could not solve the food safety.
The olive distributed and cultured in many countries except Italy. Some report suggest that the olive originated in Southwest of Asia and cultured in many countries, therefore, the 28 samples could not solve the geographical origin. Moreover, the DNA from the EVOOs was the mixed in the one place or many place and had no divergence test of local samples. Even the population was same and temperature and altitudes would affect the quality. In the 28 EVOOs had been divided into Tunisia, Italy, Spain, EU or not-EU and had no sample of other countries, which would lead the wrong conclusion.
Author Response
The EVOOs were the food and only the SSR markers could not solve the food safety.
The olive distributed and cultured in many countries except Italy. Some report suggest that the olive originated in Southwest of Asia and cultured in many countries, therefore, the 28 samples could not solve the geographical origin. Moreover, the DNA from the EVOOs was the mixed in the one place or many place and had no divergence test of local samples. Even the population was same and temperature and altitudes would affect the quality. In the 28 EVOOs had been divided into Tunisia, Italy, Spain, EU or not-EU and had no sample of other countries, which would lead the wrong conclusion.
We would thank the reviewer for the careful reading of the manuscript and considerations.
In our manuscript, solving the EVOO safety based on the use of SSR markers was not in our intent. We set up a qualitative method to attempt the identification of varieties used for the production of 28 commercial olive oils and the putative origin of olives based on the countries in which the identified varieties are mostly cultivated. Moreover, we highlighted the straightforwardness of the use of private alleles for the identification of the geographical origin of samples; however, we are aware that the list of samples used for the identification of private alleles was limited. Therefore, our future outlook is to increase that list in order to obtain a more reliable set of private alleles.
Reviewer 4 Report
Comments and Suggestions for Authors
The aim of this paper was to establish an effective method that can be used for valorization of the high-quality extra-virgin olive oil oils and their protection from fraudulent manipulations necessary for safeguarding consumer choices. The main contribution of this article is its original approach that relies on the use of molecular markers for profiling of olive oil samples. However, the presented method should be improved in order to become more reliable and cost-effective and the main constrains are suggested by authors in the discussion part.
The manuscript is clear, considering used methodology and English quality. The results are supported with very clear and illustrating tables and figures, while discussion is supported by relevant literature including the most recent publications.
The article could be improved considering the comments below:
- The introduction part is a bit wide, with some data that are not so relevant for the topic, like the part about storage conditions that starts from the line28.
- 2.2 SSR-based assay – how you chose the set of 9 SSR markers, what was the main criteria? Why you did not use more than 9 SSR markers? It is a bit low number for DNA profiling.
- Figure 3. Principal Coordinates Analysis (PCoA) performed on the 28 oil samples and the set of 149 Mediterranean olive accessions. – could you explain more this figure and the colours that you used to mark some samples.
Author Response
The aim of this paper was to establish an effective method that can be used for valorization of the high-quality extra-virgin olive oil oils and their protection from fraudulent manipulations necessary for safeguarding consumer choices. The main contribution of this article is its original approach that relies on the use of molecular markers for profiling of olive oil samples. However, the presented method should be improved in order to become more reliable and cost-effective and the main constrains are suggested by authors in the discussion part.
The manuscript is clear, considering used methodology and English quality. The results are supported with very clear and illustrating tables and figures, while discussion is supported by relevant literature including the most recent publications.
We would thank the reviewer for the careful reading of the manuscript and considerations. We addressed all the points in detail below.
The article could be improved considering the comments below:
- The introduction part is a bit wide, with some data that are not so relevant for the topic, like the part about storage conditions that starts from the line28.
We would thank for the suggestion. The Introduction section has been summarized.
- 2.2 SSR-based assay – how you chose the set of 9 SSR markers, what was the main criteria? Why you did not use more than 9 SSR markers? It is a bit low number for DNA profiling.
We clarified the criteria for selecting the 9 microsatellite markers in lines 434-436. Unfortunately, the amount of extracted DNA was insufficient to analyze a wider panel of SSR markers. However, previous studies have shown that a set of 9 microsatellite markers was able to distinguish the cultivars used for olive oil production (Testolin and Lain 2005; Vietina et al., 2011; Pasqualone et al., 2013; Piarulli et al., 2019)
- Figure 3. Principal Coordinates Analysis (PCoA) performed on the 28 oil samples and the set of 149 Mediterranean olive accessions. – could you explain more this figure and the colours that you used to mark some samples.
Following the reviewer’s suggestion, we better clarify the section concerning the results of the PCoA analysis.
Round 2
Reviewer 2 Report
Comments and Suggestions for Authors
The authors responded to all my suggestions. I have no more comments.